# ROBUST ONLINE INFERENCE USING ADAPTIVE MODEL SWITCHING

**Kalpan Mukherjee, Vikramank Singh,**
**Abishek Sankararaman, Balakrishnan Narayanaswamy, Tim Kraska**
Amazon Web Services
{kmukher, vkramas, abisanka, muralibn}@amazon.com
kraska@mit.edu

## ABSTRACT

It is well known that Large language models (LLMs) have good zero-shot and few-shot performance which makes them a promising candidate for inference when no or few training samples are available. However, when there is abundant task data, small custom trained models perform as well or are superior in performance to pre-trained LLMs, even after accounting for in-context examples. Further, smaller models are far cheaper and easier to maintain and serve for online traffic. This paper studies algorithms to optimally switch between such models for online inference. In the case when inference traffic is stationary, it makes sense to start with LLMs during the cold-start phase, and then switch over to small custom models once there is sufficient data. However, when distribution shifts are encountered, it is essential to fall back on LLMs while the custom models adapts to the distribution shift. We present an empirical study of such switching behaviors on 3 common real-world tasks like classification, regression, and forecasting across different data modalities like images, text, and time series and show how they can add value from the perspective of both cost and performance.

## 1 INTRODUCTION

Online inference suffers from two classical problems that have been studied in literature for a long time: (a) cold-start Volkovs et al. (2017); Schein et al. (2002); (b) distribution shift Gibbs & Candes (2021). Cold-start is a common problem in online tasks like outlier detection Grbovic et al. (2013); Roth et al. (2022), forecasting Jegannathan et al. (2022), and recommendation systems Lika et al. (2014); Feng et al. (2021); Xu et al. (2022); Vartak et al. (2017) where the trained models need some runway to collect data about the given task or user before making meaningful predictions. This runway can vary from few hours to several days depending on the downstream task. Distribution shift Gibbs & Candes (2021); Kim et al. (2021); Dragoi et al. (2022) typically occurs when the test samples start to diverge from the data distribution which model was initially trained on. This makes it impossible for organizations to

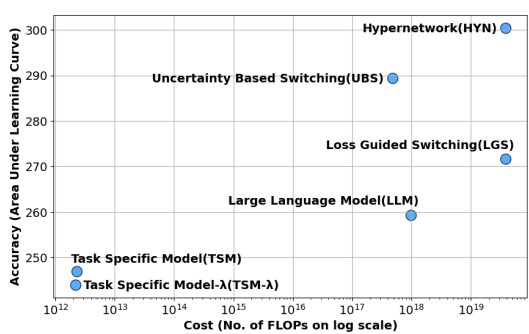

Figure 1: **Cost vs Performance trade-off.** Switching algorithms (HYN, UBS, LGS) provide more value than LLM- and TSM-only approaches under cold-start and distribution shifts.

*deploy-and-forget* models in production. But with the availability of pre-trained foundation models there's a growing interest in using them to address these two long standing problems Wu et al. (2024); Zhang et al. (2025); Horn et al. (2024); Zeng et al. (2024).

Research shows that pre-trained LLMs are reasonably good out-of-the-box on a wide range of tasks and emergent capabilities like few-shot learning or in-context learning (ICL) allow them to quickly

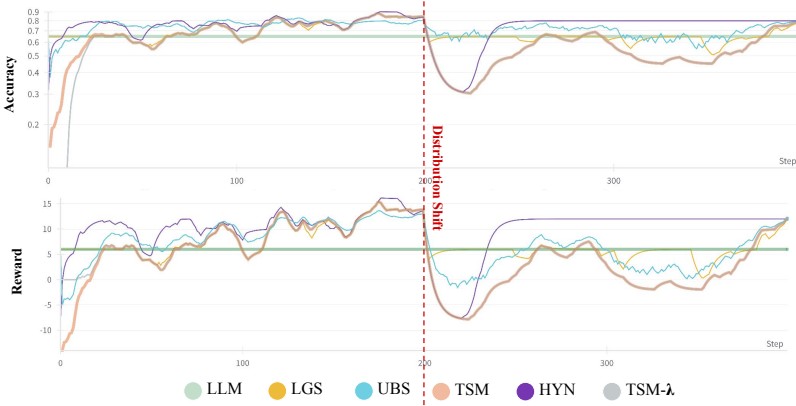

Figure 2: **Classification (MNIST/Images):** Accuracy and Reward plots evaluated on a fixed held-out test set over time. The red dotted line represents a label shift. At each time step, $N = 10$ samples arrive which the models use to learn and improve on the test-set over time as can be seen in the graph. For clear visualization we only show the best performing LLM *(in green)* chosen from different variants like zero-shot, 1-shot, few-shot, etc.

adapt to unseen tasks with a handful of examples where task-specific neural nets struggle Radford et al. (2019); Brown et al. (2020). Thus a natural question to ask is- **'RQ1: For online tasks, can pre-trained foundation models (such as LLMs) be used in-place of task-specific models (TSM) to alleviate the cold start problem?'**. Similarly, when a distribution shift occurs, a trained TSM is quickly rendered obsolete, requiring it to adapt to the new distribution. This adaptation process or re-training a TSM to meet the production requirements is time-consuming, impacting the effectiveness of downstream service. However, given LLMs few-shot learning capabilities, another interesting question is- **'RQ2: Can LLMs serve as temporary replacement during distribution shift until a new TSM is available?'**. It is also worth noting that pre-trained LLMs are exponentially costlier than small TSMs for running inference at a production scale Irugalbandara et al. (2024) and this cost grows with the number of in-context examples. Furthermore, works like Liu et al. (2024) show that LLMs are sensitive to context position, i.e., they can produce different outputs for the same input depending on the positions of in-context examples or the task instructions making them unreliable when context length grows in size. These examples shows that choice of *'right'* model isn't stationary. Thus, the final question we examine is- **'RQ3: How to identify the optimal switching points b/w TSMs and LLMs to minimize cost and maximize accuracy?'**.

We start with demonstrating the existence of phenomenons presented as RQ1 and RQ2 on multiple real-world datasets across different modalities. We then show that simple switching algorithms are sufficient to strike the necessary balance between LLMs and TSMs and achieve better cost and performance than individual LLMs and TSMs under distribution shifts. Choosing the best pre-trained LLM at inference time is a crucial problem for real-world setting Xia et al. (2024a); Huang et al. (2025); Okanovic et al. (2024) both from cost and performance standpoint. We extend this problem setup to include TSMs in the mix and study the behavior under distribution shifts which is a fundamental problem in online setting.

## 2 PROBLEM STATEMENT

Let $(X_t^i)_{i=1}^N$ be a batch of $N$ samples arriving at time $t$ and $(Y_{t+\eta}^i)_{i=1}^N$ be the ground truth labels of those samples that arrive at time $t + \eta$, where $\eta > 0$. The ground truth labels are subject to noise where $\beta\%$ of samples have incorrect labels ($\beta \geq 0$). There are $K$ different models that one can choose from to run a forward pass (a.k.a inference). At-each time $t$, one can run forward pass on at-most $1 \leq M_1 \leq K$ models and then do a backward pass or update at-most $0 \leq M_2 \leq K$. A backward pass for a pre-trained LLM model could be as simple as storing $(X^i, Y^i)$ in a running memory to be able to use it for ICL. At any time $t$, task is to pick 1 out of $K$ models to run inference for each sample in the arriving batch, with a goal of maximizing overall performance over a given inference period of length $\tau$. Let $(C_{LLM}^f, C_{LLM}^b)$ and $(C_{TSM}^f, C_{TSM}^b)$ be the cost of running a

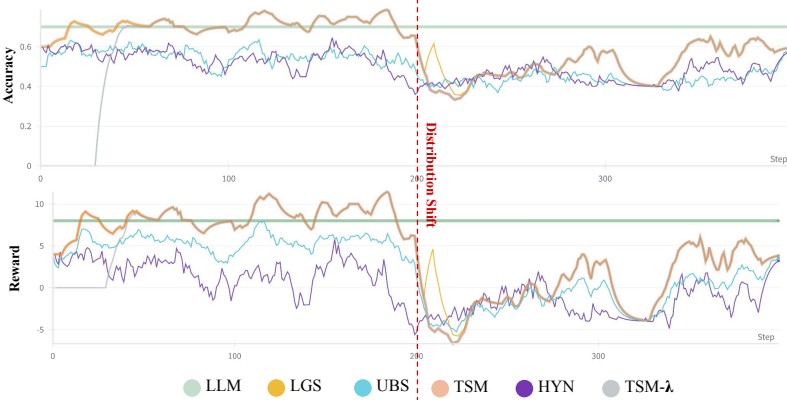

Figure 3: **Classification (Sentiment Analysis/Text)**: Accuracy and Reward plots evaluated on a fixed held-out test set over time. The red dotted line represents a label shift. At each time step, $N = 10$ samples arrive which the models use to learn and improve on the test-set over time as can be seen in the graph. For clear visualization we only show the best performing LLM *(in green)* chosen from different variants like zero-shot, 1-shot, few-shot, etc.

forward and backward pass on a LLM and TSM respectively. Similarly, let $C_{switch}$ be the switching cost between a LLM and TSM.

In this preliminary study, we consider the simplest non-trivial instance of this problem. We assume $C_{LLM}^b = C_{TSM}^b \approx 0$ and $C_{LLM}^f >> C_{TSM}^f \approx 0$. We also ignore the switching cost, i.e., $C_{switch} \approx 0$. We assume $\eta = 1$ i.e., labels arrive immediately at the next step; $K = 2, \gamma = 1$, i.e., a binary decision of model selection for each sample, and no incorrect labels, i.e., $\beta = 0$. This setting is already complex enough to capture several problem nuances as we detail in the Appendix A.5.

## 2.1 MOTIVATION

In real-world applications like online recommendation systems or anomaly detection, new customers with unique behavior arrive all the time making it harder to deploy one-size-fit-all models. To reduce false positives in classical ML a cold-start period is introduced to adapt to the behavior of new customer during which no recommendations or detections are surfaced to the customer. This period can last anywhere from few minutes (e.g., movie recommendations) to few days (e.g., fraud detection) depending on the nature and severity of the task. Similarly, seasonal changes, new releases, deprecation of old features, etc lead to unpredictable shifts in data distribution that classical TSMs are not robust to, requiring them to constantly adapt. This adaptation typically means retraining the TSM from scratch or fine-tuning last few layers on large amounts of newly collected data leading to an increase in downtime of the service. Foundation models (FMs) on other hand possess features like ICL on a small set of examples making them ideal replacement of TSMs for such downtimes. However, aside from being exponentially costlier, LLMs are seen to become less-effective with growing context length Liu et al. (2024); Tan et al. (2024). This demands an investigation into the class of solutions which can robustly switch between LLMs and TSMs depending on incoming data to provide the best cost and performance value for the downstream service.

## 3 EXPERIMENTAL SETUP

**Datasets.** We examine the 3 research questions on 2 common tasks: classification, and regression across 3 different data modalities: (1) image, (2) time series, and (3) text. These form the basis of several popular real-world use-cases like recommender systems, fault detection, stock prediction, weather analysis etc. For classification we use the MNIST dataset Deng (2012), IMDB dataset of

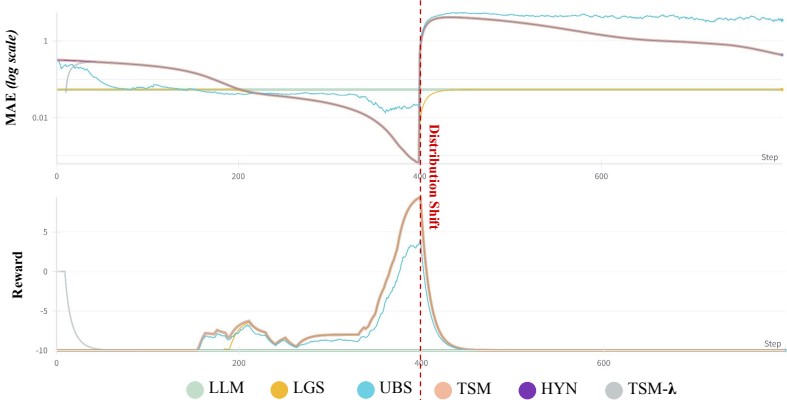

Figure 4: **Regression (Time Series)**: MAE Loss and Reward plots evaluated on a fixed held-out test set over time. The red dotted line represents a covariate shift. At each time step, $N = 10$ samples arrive which the models use to learn and improve on the test-set over time as can be seen in the graph. For the first plot, lower is better while for second, higher is better.

| Task *(Modality)* | LLM | TSM | LGS | UBS | HYN | TSM-$\lambda$ |
|---|---|---|---|---|---|---|
| Classification *(Image)* | 259.35 | 247 | 271.65 | 289.42 | **300.45** | 244.02 |
| Sentiment Analysis *(Text)* | **279.30** | 242.15 | 244.10 | 198.65 | 198.65 | 222.25 |
| Regression *(Time Series)* | 43.26 | 817.08 | **36.50** | 1785.46 | 817.08 | 814.10 |

Table 1: Comparing Area-under-learning-curve (ALC) for accuracy on test set for all the models across time on a variety of different tasks and modalities. Higher is better for classification tasks (Row 1 and 2) but lower is better for regression task (Row 3) where we report ALC for MAE curve.

movie reviews [1], and Twitter sentiment analysis dataset [2]. For the regression task we use Bitcoin Historical Dataset Kottarathil (2020). Given the likelihood that LLM must've already been trained on these public datasets, we modify the datasets by feature transformations and label obfuscations to truly test the ICL capabilities of LLMs. For e.g, in MNIST dataset we replaced the digit labels to random names like: $\{0 : \text{beta}, 1 : \text{alpha}, 3 : \text{gamma}, \ldots\}$. More details on obfuscation is provided in Appendix A.

**Models:** We use Claude 3 Sonnet model Anthropic (2024) as the pre-trained LLM and a standard 2-layer neural-net as a TSM trained with ADAM Diederik (2014) optimizer and a cross-entropy loss or a quantile loss depending on the nature of task.

**Cold-start setup:** We create an online inference setup where both LLM and TSM have 0 samples to begin with. At each time step $t$, a batch of $N = 10$ samples arrive for inference. A switching algorithm picks one of the two model for each sample in the batch and runs the inference. At next step, true labels for the batch arrives, which is then used to update the two models. For TSM, update refers to a gradient descent step, while for LLM is adding examples in context for ICL. The process is repeated in online manner. A fixed held out test set is used to report the performance at each step.

**Distribution shift setup:** Setup is similar to cold-start, however, we introduce a shift half-way through the run as shown in Figures 2, 3, and 4. For classification task this shift is a label shift, i.e., we alter the distribution of labels, and for regression we introduce a covariate shift using an exponential transformation on the incoming test samples half-way through the timeline.

**Evaluation Metrics:** We use the area-under-the-learning-curve (ALC) as our evaluation metric Gonzalez-Gutierrez et al. (2023). A higher ALC score denotes superior performance across time under both cold-start and distribution shifts. It captures the overall performance of an algorithm throughout the learning process taking into account both the speed of learning and the final accu-

---

[1]https://www.kaggle.com/datasets/lakshmi25npathi/imdb-dataset-of-50k-movie-reviews
[2]https://www.kaggle.com/datasets/jp797498e/twitter-entity-sentiment-analysis

| Task *(Modality)* | LLM | TSM | LGS | UBS | HYN | TSM-$\lambda$ |
|---|---|---|---|---|---|---|
| Classification *(Image)* | 2394 | 1900 | 2886 | 2760.5 | **4038** | 1971.0 |
| Sentiment Analysis *(Text)* | **3192.0** | 1706.0 | 1784.0 | 837.0 | $-34.0$ | 1500.0 |
| Regression *(Time Series)* | $-7990.0$ | $-6736.0$ | $-6818.0$ | $-7099.0$ | $-7990.0$ | **-6641.0** |

Table 2: Comparing Area-under-learning-curve (ALC) for reward on test set for all the models across time on a variety of different tasks and modalities. Higher is better.

racy achieved. It's particularly useful for comparing algorithms in online or incremental learning scenarios where data arrives in a stream.

We report ALC under two curves: 'accuracy vs time' (Table 1) and 'reward-vs-time' (Table 2). Here reward is computed as $-1$ for incorrect answer, $0$ for not answering, $+1$ for a correct answer. This kind of scoring is particularly useful in scenarios where False Positives (FPs) are detrimental to the downstream service (e.g., fraud detection). For regression task, we assign reward as $+1$ only if the prediction is within $\pm 10\%$ of the true value. We also compare the total cost of running these models under various switching algorithms to study how expensive these algorithms are compared to baselines (Table 3).

## 4 RQ1: CAN LLMS BE USED IN-PLACE OF TSMS TO ALLEVIATE THE COLD START PROBLEM?

The cold start period occurs in the first half i.e., the left hand side of the distribution shift line in red in all tasks– Figure 2, Figure 3, and Figure 4. The TSM starts afresh at $t = 0$ which is visible in its initially poor test accuracy, while the pre-trained LLM is clearly superior across all 3 tasks. This difference is emphasized in the reward plots where we penalize models for incorrect predictions. For e.g., in MNIST classification (Fig 2) the TSM starts from a high negative reward compared to the LLM and its cold-start variant TSM-$\lambda$ which doesn't produce any output for first $\lambda$ steps. Even on a difficult task like time series regression (Fig 4) the performance gap, although smaller, still exists. These results support the hypothesis that LLMs could serve as placeholders during the cold-start period in-place of TSM or even TSM-$\lambda$. We also hypothesized in the beginning that the cold-start period is variable and depends on the nature of the task. This too can be observed in the plots where the point in time at which TSM begins to beat LLM on test-accuracy is different across tasks depending on their complexity.

## 5 RQ2: CAN LLMS SERVE AS TEMPORARY REPLACEMENT OF TSMS DURING DISTRIBUTION SHIFT?

The red line in the plots denotes an introduction of distribution shift in form of either a label shift or a covariate shift depending upon the nature of the task. From all 3 tasks we can see the performance of TSM is severely impacted by the distribution shift and the time it takes to recover is dependent on the incoming data distribution and complexity of the task. For e.g., in MNIST classfication (Fig 2) the TSM is able successfully beat the LLM in $\sim 200$ steps after the shift. However, in more complex tasks like regression (Fig 4) or LLM-friendly task like sentiment analysis (Fig 3) it takes TSM a long time to even come close to the LLM-level accuracy after the shift. These results across multiple tasks, modalities, and types of shifts provide reasonable evidence that the drop in performance of TSMs can be alleviated by substituting it with a pre-trained LLM during a distribution shift.

## 6 RQ3: HOW TO IDENTIFY THE OPTIMAL SWITCHING POINTS B/W TSMS AND LLMS?

In this section we explore a set of simple baselines and learned methods that can achieve a higher ALC score than some trivial practices that are common in real-world. We study the following approaches:

| Task *(Modality)* | LLM | TSM | LGS | UBS | HYN | TSM-$\lambda$ |
|---|---|---|---|---|---|---|
| Classification *(Image)* | $9 \times 10^{17}$ | $2 \times 10^{12}$ | $3 \times 10^{19}$ | $4 \times 10^{17}$ | $3 \times 10^{19}$ | $2 \times 10^{12}$ |
| Sentiment Analysis *(Text)* | $7 \times 10^{17}$ | $3 \times 10^{14}$ | $3 \times 10^{20}$ | $4 \times 10^{17}$ | $3 \times 10^{20}$ | $2 \times 10^{14}$ |
| Regression *(Time Series)* | $1 \times 10^{17}$ | $7 \times 10^{12}$ | $3 \times 10^{19}$ | $1 \times 10^{17}$ | $3 \times 10^{19}$ | $7 \times 10^{12}$ |

Table 3: Comparing total cost incurred on test set by all the models across time on a variety of different tasks and modalities. Here the cost is represented as the no. of FLOPs. Lower is better.

- *(Baseline 1)* LLM: In this method we simply use a pre-trained LLM as the inference method across time. New samples with labels are added in-context for ICL as they arrive.

- *(Baseline 2)* TSM-$\lambda$: In this method an untrained TSM is used as inference method and trained on new samples as they arrive online. Here $\lambda$ denotes the cold-start period, i.e., for $\lambda$ steps the TSM will be in learning mode and hence will not perform inference. The value of $\lambda$ is usually domain dependent.

- *(Baseline 3)* Loss-guided switching (LGS): This method uses a moving window of size $T$ to compute the mean loss over last $T$ steps for each model and uses that to select the best model for next inference step. Drawback of this approach is that it runs both the models in background to compute the running loss and select the best one for inference.

- *(Learned method 1)* Hypernetwork (HYN): This method learns a binary classifier that maps the input features to the right model to choose for inference. It is trained in an online manner as new samples and labels are observed.

- *(Learned method 2)* Uncertainty-based switching (UBS): This method uses TSM's prediction uncertainty to decide when to switch over to LLM. We use entropy to measure the uncertainty of predictions Louizos & Welling (2017) and if it below a set threshold $\phi$ we deem the TSM to be confident, otherwise we use the LLM to generate the response. Similarly, for regression task we use a quantile loss to train the TSM and set a threshold on the Interquartile Range (IQR) to determine the confidence of model.

LGS, HYN, and UBS are the three simple switching algorithms with different cost vs performance trade-offs. Figure 1 provides a good overview of how these models compare on scale of both performance and cost. Table 1 shows the ALC score which represents the overall performance of models across time with periods of cold start and distribution shifts. We can see that HYN achieves the best score on classification task, even better than individual LLM and TSM models, closely followed by UBS. HYN also achieves the highest reward (Table 2) indicating lower false positive rate. HYN is a relatively costlier algorithm as it learns a mapping from features to right model and performs better if both TSM and LLM are constantly run on all samples in background.

UBS on the other hand relies heavily on TSM's uncertainty estimation and is likely to suffer on tasks that are naturally harder for TSMs. This is seen in Sentiment Analysis task where LLM is the best performant model closely followed by LGS. Since LGS relies on last $T$-step loss estimates of both models, it turns out to be a more reliable switching method for harder tasks like sentiment analysis and time series regression where LGS is the most performant, closely followed by LLM. This is visualized in Fig 4 where LGS (yellow) sticks with LLM in beginning, then switches to TSM after it is able to defeat LLM around the $t = 200$ mark and after the shift, it gradually switched back to LLM given the dip in TSM's performance.

## 7 CONCLUSION AND FUTURE WORK

Our work explores test-time switching algorithms between LLMs and TSMs to improve performance without hurting costs. We demonstrate that combining LLMs with TSMs can enhance performance, particularly during cold-start and distribution shift phases. Preliminary results show that simple switching algorithms can, in most cases, outperform both LLMs and TSMs individually across various tasks and modalities. We are currently working on designing principled algorithms that borrow from the study of non-stationary bandits Besson et al. (2022) , budgeted and combinatorial bandits Zhang & Cheung and rising bandits Metelli et al. (2022); Xia et al. (2024b) to propose a comprehensive methodology to handle the open questions highlighted in Sec A.4.

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

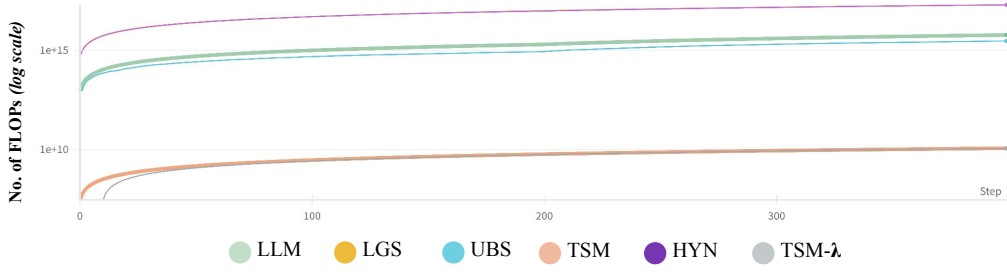

Figure 5: **Classification (MNIST/Images)**: Number of FLOPs for each model measured over time to study the cost of using each model.

## A APPENDIX

### A.1 DATA OBFUSCATION

Given the likelihood that LLM must've already been trained on almost all of the public datasets, we modify the following datasets by feature transformations and label obfuscations to truly test the ICL capabilities of LLMs. Below we discuss the strategies we adopted to transform these datasets.

**MNIST dataset.** MNIST is a primarily a classification dataset that maps from a vector representation of images to class labels. In this case, we obfuscated the labels by replacing them with random Greek letters like $\alpha, \beta, \gamma$ etc. We did this for all 10 class labels ranging from $0$ to $9$. Since there is no logical connection between the true class labels and Greek labels it is impossible for the LLM to succeed just by using its pre-trained knowledge, i.e., zero-shot inference. It should use its ICL capabilities to learn the mapping at test-time in order to perform successfully at the task.

**Twitter dataset.** In Twitter Sentiment Analysis Dataset, the task is to judge the sentiment of the message about the entity. There are three classes in this dataset: Positive, Negative and Neutral. The messages that are not relevant to the entity (i.e. Irrelevant) are regarded as Neutral. Similar to MNIST, we perform label obfuscation and replace the *'positive', 'negative'* and *'neutral'* labels with random colors like *'green', 'blue',* and *'yellow'*. This breaks the relationship between tweets and the sentiment labels that LLM might've learned during pre-training. It will need to possess ICL capabilities at test-time to generate correct obfuscated labels.

**IMDB dataset.** IMDB dataset has 50K movie reviews for natural language processing or text analytics tasks. This is a dataset mainly for binary sentiment classification containing a set of 25,000 highly polar movie reviews for training and 25,000 for testing. We perform obfuscation similar to the Twitter dataset given the similar nature of task.

**Bitcoin Historical Dataset.** Bitcoin Historical Dataset is a multivariate time series dataset where the goal is to predict the closing price from a set of categorical and continuous variables. Given the outcome variable is continuous, this dataset is typically posed as a regression task. For this dataset we perform non-linear feature transformation to modify its original form. More specifically, we perform a $\log$ transformation on the dataset which modifies it in a way where memorization skills would no longer be of use. The LLM will need to use its ICL capabilities to learn the right relationship between features and bitcoin prices at test-time to perform well on the task.

### A.2 RESULTS (COST)

In Table 3 we compare the total runtime cost of all the models across tasks. It is interesting to see how close some of the switching models are to the most cost effective option– TSMs and least cost-effective option– LLMs. We also show how the cost varies with time in Figure 5, Figure 6, and Figure 7. As expected, TSMs are the most cost-effective approach given their cheap inference cost compared to LLMs. HYN is the costliest model to run since it requires both TSM and LLM to be run in the background. UBS is an interesting candidate that is cheaper than LLM-only but costlier than TSM-only approach as seen in Figure 1 and across all the cost plots.

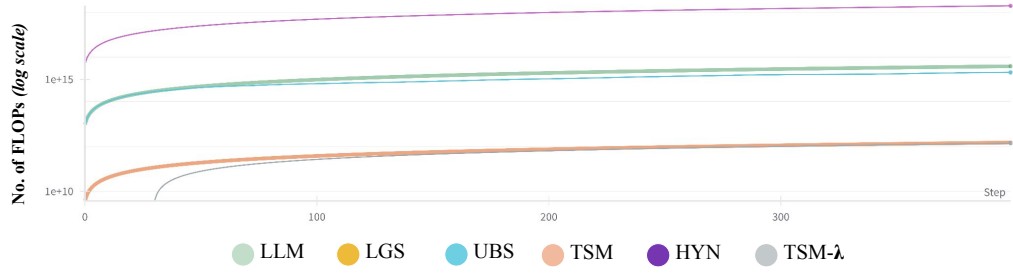

Figure 6: **Classification (Sentiment Analysis/Text)**: Number of FLOPs for each model measured over time to study the cost of using each model.

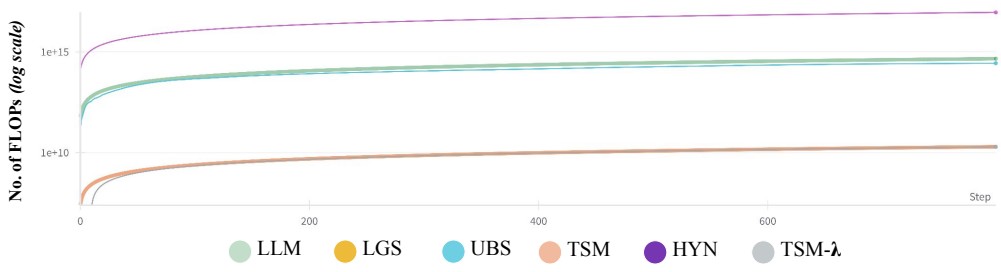

Figure 7: **Regression (Time Series)**: Number of FLOPs for each model measured over time to study the cost of using each model.

### A.3 HOW DO LLMS MAKE MODEL-SELECTION DIFFERENT COMPARED TO 'CLASSICAL ML'?

LLMs have shown to be great at zero/few shot learning, where even with very small dataset, the inference accuracy is high for many tasks. This is accomplished through a variety of methods - for eg., prompting and ICL. On the other hand, when there is abundant task data available, it has been shown that much cheaper specialized 'classical ML' models such as small neural-nets have comparable or better accuracy than LLMs prompted using ICL and/or RAG Liu et al. (2024); Tan et al. (2024). In other words, a larger model (such as LLM) is optimal for inference in the small data regime and a smaller specialized model is optimal in the large data regime. This is a reversal from conventional ML practice that in the small data regime smaller model-sizes yield better generalization as they avoid over-fitting and in the large sample regime, more complex models yield better generalization as they overcome the high bias problem of small models.

### A.4 DISCUSSION OF OPEN RESEARCH PROBLEMS

As stated in Section 2, for the scope of this paper we consider the simplest non-trivial instance of this problem. In this section we discuss the harder questions that will make the problem more complicated and closer to real-life scenarios. Goal of this section is to raise important and interesting open research questions that will intrigue researchers in this community to dive deeper in this space of problems.

**Q1. Unclear how to incorporate cost and accuracy in a single optimization function?**: Figure 1 shows a summary of how various switching models perform on both performance and cost. However, when training these models we still use the performance-driven traditional objective function like minimizing some kind of loss like MSE, cross-entropy, MAE, etc. In real-world customers always look at both cost and performance together when making business decision. This demands a custom metric which can robustly put cost and performance on a single scale. Optimization algo-

rithms that can consider both cost and performance while learning, and make trade-offs accordingly, will tend to be far more superior than algorithms presented in our work so far.

**How does this framework change when $\eta > 1$?** In real-world customers rarely provide instant feedback (i.e., $\eta = 0$ $or$ $1$). Usually the feedback is either delayed or absent making the learning process hard. In this work we assume $\eta = 1$ which makes the problem simpler but how do these learning algorithms translate to delayed feedback is an interesting question we leave for future work.

**How does this framework change when $\beta > 0$?** Another related question is the correctness of feedback. We assume $100\%$ correctness of feedback labels which is a rare scenario. In real world, labels are not only absent but in many cases incorrect which could significantly impact the leaning based switching models.

**How does this framework change when $C_{switch} > 0$?** Switching methods like UBS, LGS and HYN work well because they try to pick the best model for each sample in the arriving batch. This works well on the performance scale but is reasonable on cost scale only if we assume $C_{switch} = 0$, i.e., we can easily switch between TSMs and LLMs as needed. In reality this may not always be true as there might be infrastructure cost, delay, and other factors impacting the seamless transition between two inference models.

**How sensitive is this framework to varying user preferences (e.g., cost-sensitive vs performance-driven?)** Customers' preferences are non-stationary. They can vary from requiring the inference to be extremely performant (absolutely no tolerance for false positives and negatives) to cost sensitive (high tolerance for false alarms). This could be driven by multiple factors like nature of the task, market fluctuations, feature changes, or personal preferences. The switching algorithms should be sensitive to such customer preferences and pick models that meet the customer requirements on both performance and cost scales over time.

## A.5 Technical challenges in algorithm design

The main challenge is that similar to prior works Xia et al. (2024b), the model selection problem is an instance of the *rising bandit* problem. Roughly, the performance of a choice of a model at a given time, depends on the number of past samples it was trained on. If a given model was selected to play more in the past, then its average reward for the input at the given time is higher. This already poses significant complexities to the standard explore-exploit trade-off as highlighted in Xia et al. (2024b) and (Metelli et al., 2022).

In our setting, this challenge is exacerbated by the possibility of abrupt distribution changes. These distribution changes implies that the average performance of a model at a given time is dependent on the number of times the model was played in the past *since the distribution shift*, with higher average performance, if a model was played more since the last distribution shift. This implies that the ideal algorithm must be adaptive to not only the gradual changes that are occurring as models are selected due to one-step gradient updates or addition of a small corpus of ICL examples, but also be adaptive to distribution shifts that can make a large change in the average performance of one or more models. Further, this adaptation needs to be done online without having predictable information on when the change occurs.

## A.6 Algorithms

In this section we outline the pseudocode for the two learning based methods for switching models at test-time.

---

**Algorithm 1** Hypernetwork (HYN)

---
1: **Input:**
2:    Stream of input data $X^i$ in a batch of size $N$ at time $t$: $\{X_t^i\}_{i=1}^N$
3:    Stream of ground truth labels of the batch arriving at time $t + \eta$: $\{Y_t^i\}_{i=1}^N$
4:    Let $\eta = 1$ (time delay in arrival of labels)
5:    Let $f(\cdot)$ and $g(\cdot)$ be the update functions for LLM and TSM respectively.
6:    **Initialize** a pre-trained LLM $M_{LLM}(\cdot)$ and an untrained TSM $M_{TSM}(\cdot)$
7:    **Initialize** a binary classification model $H(\cdot)$ with parameters $\theta$.
8:    Let $\alpha$ be the learning rate and $L(\cdot)$ be a supervised cross entropy loss function.
9: **for** Time $t = 1, 2, \ldots$ **do**
10:    // Obtain predictions from each model for the previous batch
11:    $M_{LLM}(\{X_{t-1}^i\}_{i=1}^N) = \{\hat{Y}_{t,LLM}^i\}_{i=1}^N$
12:    $M_{TSM}(\{X_{t-1}^i\}_{i=1}^N) = \{\hat{Y}_{t,TSM}^i\}_{i=1}^N$
13:    Receive the true labels of previous batch at this time step, $\{Y_t^i\}_{i=1}^N$.
14:    // Identify the correct model and build a training batch
15:    $D_t = \{X^i, Z^i\}_{i=1}^N$ where $Z^i = \left[\mathbb{I}(\hat{Y}_{LLM} = Y^i), \mathbb{I}(\hat{Y}_{TSM} = Y^i)\right]$
16:    // Train the hypernetwork
17:    Compute a supervised loss $L(\theta)$ over training batch $D_t$
18:    Update the model parameters $\theta \leftarrow \theta - \alpha\nabla_\theta L(\theta)$
19:    // Update both LLM and TSM
20:    $M_{LLM}^* = f(M_{LLM}, \{X_{t-1}^i, Y_t^i\}_{i=1}^N)$
21:    $M_{LLM} \leftarrow M_{LLM}^*$
22:    $M_{TSM}^* = g(M_{TSM}, \{X_{t-1}^i, Y_t^i\}_{i=1}^N)$
23:    $M_{TSM} \leftarrow M_{TSM}^*$
24: **end for**

---

**Algorithm 2** Uncertainty based switching (UBS)

---
1: **Input:**
2:    Stream of input data $X^i$ in a batch of size $N$ at time $t$: $\{X_t^i\}_{i=1}^N$
3:    Stream of ground truth labels of the batch arriving at time $t + \eta$: $\{Y_t^i\}_{i=1}^N$
4:    Let $\eta = 1$ (time delay in arrival of labels)
5:    Let $f(\cdot)$ and $g(\cdot)$ be the update functions for LLM and TSM respectively.
6:    Let $H_t(\cdot)$ be the decision function of TSM at time $t$, i.e., entropy of predicted distribution for classification, or IQR for regression.
7:    Let $\hat{p}_t^i$ denote the prediction probability of a sample $X^i$ at time $t$ if the task is classification and $\hat{q}_t^i$ denote the predicted quantiles if the task is regression.
8:    Let $\phi$ be the decision threshold, either on entropy score or IQR.
9:    Let $D_t(\cdot)$ be the model chosen for inference at time $t$.
10:    **Initialize** a pre-trained LLM $M_{LLM}(\cdot)$ and an untrained TSM $M_{TSM}(\cdot)$
11: **for** Time $t = 1, 2, \ldots$ **do**
12:    // For the current batch of samples, obtain either the
       prediction probabilities or the quantiles from TSM
13:    $M_{TSM}(\{X_t^i\}_{i=1}^N) = \{\hat{p}_t^i\}_{i=1}^N$ or $M_{TSM}(\{X_t^i\}_{i=1}^N) = \{\hat{q}_t^i\}_{i=1}^N$
14:    **if** $H(\{\hat{p}_t^i\}_{i=1}^N) < \phi$ **then**
15:       // Use TSM for inference
16:       $D_t \leftarrow M_{TSM}$
17:    **else**
18:       // Use LLM for inference
19:       $D_t \leftarrow M_{LLM}$
20:    **end if**
21:    // Update both LLM and TSM
22:    $M_{LLM}^* = f(M_{LLM}, \{X_{t-1}^i, Y_t^i\}_{i=1}^N)$
23:    $M_{LLM} \leftarrow M_{LLM}^*$
24:    $M_{TSM}^* = g(M_{TSM}, \{X_{t-1}^i, Y_t^i\}_{i=1}^N)$
25:    $M_{TSM} \leftarrow M_{TSM}^*$
26: **end for**

---

