# OpenReview forum: "ROBUST ONLINE INFERENCE USING ADAPTIVE MODEL SWITCHING"
_ICLR.cc/2025/Workshop/MCDC — MCDC @ ICLR 2025_

### Official Review · Reviewer_MSq3 · 2025-03-02

**Rating:** 6
**Confidence:** 4
**Fit:** 4

**Summary:**

This work addresses the problem of adaptive model switching between large language models (LLMs) and smaller task-specific models (TSMs) for online inference. The motivation comes from the observation that LLMs perform well in zero-shot/few-shot settings but are generally expensive, whereas custom TSMs are more cost-effective but struggle with cold-start and distribution shifts. To balance these trade-offs, the paper proposes several adaptive switching algorithms - including Loss-Guided Switching (LGS), a Hypernetwork-based approach (HYN), and Uncertainty-Based Switching (UBS) - that decide, at each inference step, which model to use, aiming to minimize cost while maintaining accuracy. They conduct experiments across classification, regression, and forecasting tasks, using datasets from different modalities (text, images, time-series). Their results show that simple switching heuristics and learned policies can significantly outperform static strategies (always using LLM or TSM) in both performance (area under the learning curve, ALC) and cost efficiency (FLOPs).

**Reason For Giving A Higher Score:**

- The problem is relevant and well motivated.
- The work makes a contribution to the field discussing relevant derivative problems.
- The work chooses good datasets for evaluation, covering different tasks and modalities.

**Reason For Giving A Lower Score:**

- The work lacks novelty beyond empirical analysis of switching strategies.
- Its simplifying assumptions (immediate labels, no switching cost) reduce real-world applicability. The work lacks discussion on long-term model drift and gradual shifts.
- There is limited comparison with advanced model selection techniques.

**Strengths And Weaknesses:**

Strengths
1. The paper tackles a practical and important problem in online inference by leveraging the strengths of both LLMs and TSMs. The motivation is well explained, particularly the challenges of cold-start and distribution shifts.
2. The experiments cover diverse tasks and data modalities, making the findings more generalizable across different scenarios.
3. The work raises some open research questions, which can help guide future work.

Weakness
1. While the problem is relevant, the work lacks novelty since it mainly applies simple existing methods to this problem rather than introducing new techniques.
2. The experimental methodology is not fully explained, making it difficult to interpret the results. Some important details for reproducibility are missing. Additionally, the paper treats some aspects as trivial, such as using an LLM for time-series prediction, which is still an open research challenge.
3. The study focuses on an idealized scenario where distribution shifts are sudden and clearly defined. While this can be valid simplification for the scope of this work, the paper does not discuss how its findings generalize to real-world cases, where shifts are often gradual and unpredictable. It would be stronger if it drew clearer parallels between the idealized setup and real-world challenges.

**Suggestions:**

- For reproducibility, it would be helpful to release the exact in-context prompts used for LLMs in the experiments in each of the datasets/modalities.
- The paper could discuss how LLMs are used in online learning and how to adapt them for regression tasks, which would add clarity.
- It is unclear why LLM performance remains constant if new in-context learning examples are added during the experiment (as mentioned in line 267). As the prompt is changing over time, some explanation is needed.
- Why not evaluate switching back to TSMs after some time instead of keeping LLMs indefinitely after a shift? This would significantly reduce costs and align better with adaptive online learning / model selection technique. Using LLMs forever after is as prohibitively expensive as using them all the time.
- The paper's structure could be improved. Some results appear before the baselines and models are introduced, making it harder to follow. While I understand the baselines are mainly relevant for RQ3, it initially makes it difficult to understand the setup.
- The work could explore simple but more effective approaches, such as online reinforcement learning-based policies that actively explore and exploit model choices, and could optimize cost and accuracy together.
- The work could discuss more the limitations of the work scope. For example, using FLOPs as the main cost metric does not capture important factors like latency, memory usage, or energy consumption, which are critical in real-world deployment.

---

### Official Review · Reviewer_zGy8 · 2025-03-03

**Rating:** 6
**Confidence:** 2
**Fit:** 4

**Summary:**

This paper explores the problem of adaptive model switching in online inference for classification and regression tasks. The study compares the performance and cost trade-offs between using large language models (LLMs) and task-specific models (TSMs), particularly under cold-start conditions and distribution shifts. The authors conduct experiments on MNIST (image classification), sentiment analysis (text classification), and time series forecasting, demonstrating the advantages and limitations of LLMs versus TSMs. Additionally, they propose and evaluate switching methods to dynamically select between models to balance cost and performance.

**Reason For Giving A Higher Score:**

see Strengths And Weaknesses

**Reason For Giving A Lower Score:**

see Strengths And Weaknesses

**Strengths And Weaknesses:**

Strengths:

The problem of cost & performance optimization by proper model selection is a relevant topic.
The paper is easy to follow.

Weaknesses:

I am concerned by a potential lack of novelty.
The first two experimental questions (RQ1 and RQ2) are showcasing experiments to measure the existence of well known behaviors: pre-trained LLMs can be used for downstream tasks, and they can be better than specialized models, especially under low-data settings (i.e. cold-start or distribution shift).

RQ3 is the most interesting experimental question: how to design a good switching/routing mechanism to get the best of both worlds ? Here a few simple methods are proposed and tested. I am not expert enough to assess the novelty of these approaches, but it seems that SOTA routing systems are not considered, or at least the paper is lacking a proper positioning among existing solutions.



Minor:

l.302 "HYN is a relatively costlier algorithm as it learns a mapping from features to right model and performs better
if both TSM and LLM are constantly run on all samples in background." --> what is being shown in the figure then ? HYN with LLM and TSM inference at all time, or HYN with single inference type per sample ? By looking at Figure 1 I guess it is the former.

**Suggestions:**

Reduce emphasis on RQ1 and RQ2.
Better position your proposed switching/routing solutions wrt to state of the art.

---

### Official Review · Reviewer_VRUF · 2025-03-03

**Rating:** 7
**Confidence:** 3
**Fit:** 4

**Summary:**

This paper examines the potential for switching between small custom models and large pretrained language models. The goal is to solve the long-standing problems of cold start and distribution shift that plague the online inference tasks commonly encountered in machine learning.

The authors demonstrate the problematic cold start and distributional shift phenomena, then outline simple switching strategies. They find that LLMs could serve as effective placeholders during the cold-start and distribution shift periods in place of task-specific models.

**Reason For Giving A Higher Score:**

The work is important, and valuable to this workshop. Interesting future work is envisioned based on principled methods.

**Reason For Giving A Lower Score:**

The overall paper presentation is lacking. More effort on presentation should help raise the score.

**Strengths And Weaknesses:**

### Strengths
The ideas presented in this paper are appealing and should benefit the machine learning community at large.
The methodological approach is sound and well presented. For instance, the obfuscation of suspected LLM training datasets is very important to ensure fair comparison of these online inference mechanisms.


### Weaknesses
The main weakness I find with this paper is its presentation. Figures and Tables are referenced very far from where they were presented. For instance, Figure 1 is referenced at the end of the paper, even though it is crucial, particularly for introducing the various baselines. (Also, its caption is quite confusing, and should clarify that the value added here is the performance!)

The various baselines should be briefly described before encountering them in the tables. For instance, TSM-λ is first introduced at line 239. And the fact that the contribution methods are only described at the end even though they appear throughout is not conducive to clear understanding.

Some questions remain unanswered:
- Why are some values in bold in Table 2? What do they represent?
- (Line 287): How are the labels used to train the binary classifier for HYN obtained?

**Suggestions:**

The main suggestion is to address the questions I pointed out in Weaknesses. Additionally, what is $\gamma$ at line 131?

Fix minor weaknesses and typos. For instance, line 045 is missing some terms, which obfuscate understanding.

---

### Official Review · Reviewer_uqoV · 2025-03-03

**Rating:** 8
**Confidence:** 4
**Fit:** 5

**Summary:**

This paper studies online inference, where the distribution of input data undergoes some change. The paper considers two types of solutions: either using a generic large language model (LLM), using in-context-learning to make it perform well on the task at hand without any retraining, or training a small Task Specific Model (TSM) on the fly on the task at hand. Using the LLM yields out of the box good performance, at no training cost. When sufficiently enough data has been seen, the TSM becomes a specialist and becomes better than the LLM.
The authors experiments in this setting on three different domains, where the task swithces mid-training.
The authors then discuss possible switching strategies, in order to get the best of both worlds: when the task switches, first using the LLM, and then the TSM whne enough data has been gathered.
The authors focus the discussion on the training costs, assuming that the bottleneck is the cost of backprop -- hence only the TSM pays a non-zero price.

**Reason For Giving A Higher Score:**

Not sure what this means

**Reason For Giving A Lower Score:**

Not sure what this means

**Strengths And Weaknesses:**

# Strengths
- the paper is well written, the exposition is very clear
- the problem studied by the authors is a very important problem for the deployment in the wild of ml models, in the paradigm of large pretrained models.
- The experiments are convincing
- The discussion is very interesting.

# Weaknesses
- The problem studied here has many different parameters, like TSM and LLM scale, LLM performance, type of distribution shift, hardness of the task to address, etc. There are also different types of relevant constraints, on either the cost of backward or forward, or cost of switching models. It is hard to explore this space exhaustively, and the authors already do a good job at clarifying the picture. However, I think that exploring more the direction of model scale would greatly enhance the paper. Indeed, if the TSM is small enough, the bottleneck is no longer the cost of backprop through it, but rather the inference of the LLM. The fact that the trained TSM is better than the LLM is also very task and TSM scale dependent (for instance, on a very hard reasoning task, a TSM trained for ages on the task cannot outperform the LLM). I think that having a more thorough discussion about these points would make the paper better

- likewise, the assumption that we only pay the cost of backprop is a strong choice, which restricts quite a lot the generality of the paper's discussion (indeed, automatic differentiation theory guarantees that cost of backprop < 3 * cost of inference, and in many cases I expect that cost of backprop through TSM << cost of inference for LLM). Also, in many applications, one only trains a few models but these models are served many times; in that case the cost of inference dominates.

**Suggestions:**

I think that discussing the points mentioned above would clarify the paper.

---

### Official Review · Reviewer_3WD4 · 2025-03-03

**Rating:** 6
**Confidence:** 5
**Fit:** 4

**Summary:**

The paper studies when a pre-trained LLM should be used for a particular task as opposed to a task-specific trained model (TSM) in the online setting where a constant stream of observations and labels are observed. The authors study this question predominantly from the lens of the cold-start problem as well as distribution shifts and uncover that leveraging an LLM with in-context examples is beneficial for the cold-start problem (i.e. in low task-specific data regime) as well as in the case of distribution shifts. Overall, they pose an interesting question and show analysis across tasks on different modalities.

**Reason For Giving A Higher Score:**

The problem formulation fits well with the workshop theme and the work tackles an interesting and relevant problem. The analysis is interesting and highlights clear cases when and where TSM might outperform an LLM, and when not.

**Reason For Giving A Lower Score:**

The analysis done is a bit preliminary and does not handle a number of interesting cases: eg. TSMs as finetuned LLM models or finetuning done on distilled LLM models. It is expected that LLMs would outperform TSMs in the cold-start problem **because** they are pre-trained on language while TSMs have no signal in the start. It is also expected they will do well on distribution shifts **because** they have been trained on a large corpora of a number of different distribution shifts. In this regard, the findings are not surprising.

**Strengths And Weaknesses:**

**Strengths**

- The tasks considered in the paper are fairly diverse and they show that the conclusions the authors arrive at are shared across different modalities.
- The problem studied in the work is quite interesting and relevant, especially from the lens of production.

**Weaknesses**

- The overall writing and presentation of the work could be significantly improved (see suggestions).
- The experimental details are missing. When the authors put examples in the context of the LLMs, how do they handle increasing number of tokens?
- Do their experimental results hold in the case of a different LLM or a different TSM? The authors should provide some form of sensitivity analysis on models as well.
- While relevant, why don't the authors consider the more natural setting where the TSM is a fine-tuning of an LLM model on the observations received?

**Suggestions:**

- The writing can be significantly improved. In particular, I found that the description of the general problem misleading and unnecessary given the authors only tackle a particular setting of it. While I agree that the setting they studied is non-trivial, it could have just been explained as is without the added complexity of $\beta$ or multiple models, which I believe should only be there if the authors are showing some preliminary analysis on it.
- The authors talk about TSM-$\lambda$ before it has even been introduced. They should consider re-ordering some parts of the draft.
- The plots are a bit unreadable. The authors should consider a different color-scheme, line-width and introducing some markers.

---

### Decision · Program_Chairs · 2025-03-06

**Decision:**

Accept

**Comment:**

This work proposes how to switch between multiple models in inference. This is relevant to this workshop, as how to exploit an ensemble of networks is a form of modularity. The reviewers recommend acceptance, and we're happy to accept it to the workshop.